# The Role of Nitrogen Fertilization on the Occurrence of Regulated, Modified and Emerging Mycotoxins and Fungal Metabolites in Maize Kernels

**DOI:** 10.3390/toxins14070448

**Published:** 2022-06-30

**Authors:** Valentina Scarpino, Michael Sulyok, Rudolf Krska, Amedeo Reyneri, Massimo Blandino

**Affiliations:** 1Department of Agricultural, Forest and Food Sciences (DISAFA), Università degli Studi di Torino, Largo Braccini 2, 10095 Grugliasco, TO, Italy; valentina.scarpino@unito.it (V.S.); amedeo.reyneri@unito.it (A.R.); 2Center for Analytical Chemistry, Department of Agrobiotechnology (IFA-Tulln), University of Natural Resources and Life Sciences, Vienna (BOKU), Konrad-Lorenz-Str. 20, 3430 Tulln, Austria; michael.sulyok@boku.ac.at (M.S.); rudolf.krska@boku.ac.at (R.K.)

**Keywords:** aurofusarin, beauvericin, bikaverin, culmorin, deoxynivalenol, deoxynivalenol-3-glucoside, fumonisins, fusaric acid, moniliformin, zearalenone

## Abstract

The European Food Safety Authority is currently evaluating the risks related to the presence of emerging mycotoxins in food and feeds. The aim of this study was to investigate the role of soil fertility, resulting from different nitrogen fertilization rates, on the contamination of regulated mycotoxins and emerging fungal metabolites in maize grains. The trial was carried out in the 2012–2013 growing seasons as part of a long-term (20-year) experimental platform area in North-West Italy, where five different N rates, ranging from 0 to 400 kg N ha^−1^, were applied to maize each year. Maize samples were analyzed by means of a dilute-and-shoot multi-mycotoxin LC-MS/MS method, and more than 25 of the most abundant mycotoxins and fungal metabolites were detected. Contamination by fumonisins and other fungal metabolites produced by *Fusarium* spp. of the section *Liseola* was observed to have increased in soils that showed a poor fertility status. On the other hand, an overload of nitrogen fertilization was generally associated with higher deoxynivalenol and zearalenone contamination in maize kernels, as well as a higher risk of other fungal metabolites produced by *Fusarium* spp. sections *Discolor* and *Roseum*. A balanced application of N fertilizer, in accordance with maize uptake, generally appears to be the best solution to guarantee an overall lower contamination by regulated mycotoxins and emerging fungal metabolites.

## 1. Introduction

Mycotoxin contamination of agricultural crops is a serious threat to human and animal health, due to their acute and chronic toxicity [1]. The Food and Agricultural Organization (FAO) has suggested that about 25% of crops throughout the world are contaminated by mycotoxins [2]. Among all agricultural commodities, maize is one of the crops that is the most susceptible to and threatened by mycotoxin contamination, both in the field and during storage, due to the colonization of different fungal species which can coexist and influence each other during growth and lead to the production of multi-mycotoxins in the field [3].

Although maize is mainly subject to contamination by fumonisins B (FBs), aflatoxins (AFs), zearalenone (ZEA) and deoxynivalenol (DON), which are the mycotoxins that have been reported and monitored the most [4] due to their toxicity and to the maximum regulatory limits set in Europe by the European Commission (EC) [5,6], approximately 400 mycotoxins are known to date [7]. The mycotoxins and fungal metabolites not subject to regulation are commonly referred to as “novel” or “emerging” mycotoxins [8], and the European Food Safety Authority (EFSA) has recently issued a scientific opinion on the risks to public health concerning their presence in food and feeds [9,10]. Although the toxicological effects of these compounds have been mentioned in the literature, some of them could be toxic for humans and livestock, and the main risks, due to their co-presence, could be related to additive or synergistic effects, if they are present together with regulated mycotoxins.

In such a framework, appropriate field management and preparation can be particularly relevant for the control of mycotoxins, as indicated in Commission Recommendation 2006/583/EC on the prevention and reduction of *Fusarium* toxins in cereals and cereal products. Many agricultural practices, including the avoidance of stress induced by a lack of water or unbalanced nutrients [11], have an impact on the prevention and control of mycotoxins. 

Moreover, because of rising domestic and global market demands, one of the main objectives of maize growers is to meet the challenge of European safety quality standards, which means that everyone involved in the production chain should strive to minimize risks and, at the same time, obtain safe raw material with high nutritional quality [12]. For this purpose, soil fertility, which may be responsible for low productivity in areas destined for both grain and fodder and can also impact the safety of maize crops, is one of the main factors that should be evaluated and controlled. This necessity is not only due to low levels of nutrients in soil, but also to the inappropriate use of fertilizers, particularly nitrogen (N) and potassium, and to the high extraction capacity of maize crops [13].

Although N is essential for plants, it interferes with several features related to the growth and development of the plant, which directly or indirectly affect the crop yield and quality. Information on the influence of N fertilizer on maize grain quality and safety is still lacking and has only been reported for the regulated mycotoxins by a few authors, while conflicting effects on FBs have sometimes been forwarded. However, Blandino et al. [14] and Souza et al. [13] reported that an increase in FB contamination is mainly related to N deficiencies, while an increase in ZEA content is due to a high N fertilizer application (> 300 kg N ha^−1^). On the other hand, Marocco et al. [15], Ariño et al. [12] and Bocianowski et al. [16] reported that the use of N fertilization tended to increase the FB content of maize and increase *Fusarium verticillioides* infection [13].

To the best of the authors’ knowledge, no previous work has investigated the effect of N fertilization on the contamination of maize by emerging mycotoxins and fungal metabolites. The objective of the present work is to investigate the effect of the N fertilizer rate (0, 100, 200, 300 or 400 kg ha^−1^) on the susceptibility of maize to ear rot and to multi-mycotoxin contamination, due to both regulated and emerging mycotoxins and fungal metabolites, in naturally infected field trials conducted over two consecutive years (2012 and 2013) in North-West Italy.

## 2. Results

### 2.1. Metereological Trends

Different rainfall and temperature (expressed as growing degree days, GDDs) trends were recorded during the two growing seasons (2012 and 2013) (Table 1).

The 2012 and 2013 years were characterized by similar GDDs, although 2012 experienced drier conditions, particularly during ripening. The most rainfall was recorded in 2013 during the overall growing season; although, as far as both rainfall and rainy days are concerned, they were more concentrated in July than in 2012. The lower amount of rainfall that occurred in the summer months (June–August) in the year 2012 and the drier conditions that occurred during ripening also resulted in an earlier harvest date in 2012 than in 2013.

### 2.2. Grain Yield

As far as grain yield is concerned, the interaction between year and N rate (kg ha^−1^) was always significant, and the two years were therefore considered separately. Grain yield was, on average, higher in 2013 than in 2012, although greater increases in grain yields were recorded when switching from a low to a high N rate for the 2012 year than for 2013. The higher N rates (200, 300 and 400 kg N ha^−1^), on average, increased the grain yield significantly by 55% and 22% in the 2012 and 2013 years, respectively, compared to the lower N inputs (0 and 100 kg N ha^−1^) (Table 2). When considering the two extreme values of the N rate, it was possible to observe that there was a significant increase in the grain yield, that is, 96% and 36% in the 2012 and 2013 years, respectively, moving from 0 to 400 kg N ha^−1^; although a balanced N rate of 200 kg N ha^−1^ allowed a significant increase in the grain yield, that is, 73% and 33% in the 2012 and 2013 years, respectively, compared to a zero N rate.

### 2.3. Fungal Ear Rot and European Corn Borer (ECB) Symptoms

The interaction between year and N rate (kg ha^−1^) was significant, as far the fungal ear rot and ECB symptoms are concerned, and, consequently, the two years were considered separately. The fungal ear rot and ECB symptoms, on average, were much higher in 2012 than in 2013 (Table 2). The ear rot and European Corn Borer (ECB) incidences were never affected by the N rate, in either 2012 or 2013, while the severities of the same parameters were significantly higher for a zero N rate (+221% and +80% for ear rot and ECB severity, respectively) than the mean for any other N input considered in the trial, albeit only for the 2012 year.

### 2.4. FBs and the Related Emerging Mycotoxins and Fungal Metabolites Produced by Fusarium spp. Section Liseola 

The contamination by FBs and the related emerging mycotoxins and fungal metabolites produced by *Fusarium* spp. section *Liseola* was considered separately for the two years, due to the significant interaction between year and N rate.

On average, the FB contamination level (Figure 1) was much higher in 2012 (18470 µg kg^−1^) than in 2013 (377 µg kg^–1^). When the averages of the different N rates were considered, FB_1_ accounted about for 60% of the FBs, FB_2_ for 19% and FB_3_ for 21% in 2012, while FB4 was not detected. On the other hand, FB_1_ was about 75% of the FBs in 2013, FB_2_ was 20%, FB_3_ was 2% and FB_4_ was 3%. A significant increase (+134%) in the FB contamination level was only recorded in 2012 for zero N input, compared to the other N rates, while no differences were recorded in 2013.

A similar behavior was observed for the emerging mycotoxins and fungal metabolites detected in the maize grains (beauvericin = BEA, bikaverin = BIK, fusaric acid = FA, fusaproliferin = FUS, fusarin C and moniliformin = MON) produced by the same *Fusarium* spp. section *Liseola* that produces FBs (Table 3). Indeed, FA and FUS showed a significant increase (+384% and +116% for FA and FUS, respectively) for the zero N rate in comparison to the other N rates, albeit only in 2012. MON, instead, showed a significantly higher contamination level in 2012 for the zero N rate (+83%) than for the N rate of 200 kg N ha^−1^. The zero N rate for BIK, like MON, showed a significantly higher content in 2012 than the N rates of 200 and 300 kg N ha^−1^ (+169% and +108% for 200 and 300 kg N ha^−1^, respectively). Conversely, BEA did not show any significant differences between the N rates, although higher levels of contamination were recorded for the zero N rate than for the other rates, with the exception of 400 kg N ha^−1^. No significant difference was observed between the different N rates for any of the previously reported emerging mycotoxins or fungal metabolites in 2013. A higher contamination level was recorded for all these mycotoxins in 2012 than in 2013, in the same way as for the FBs.

### 2.5. DON, ZEA and the Related Emerging Mycotoxins and Fungal Metabolites Produced by Fusarium spp. Sections Discolor, Roseum and Sporotrichiella 

Contrary to what was observed for the FBs and the related emerging mycotoxins and fungal metabolites produced by *Fusarium* spp. section *Liseola*, DON, ZEA and the related emerging mycotoxins and fungal metabolites produced by *Fusarium* spp. sections *Discolor*, *Roseum* and *Sporotrichiella* showed higher contamination levels in 2013 than in 2012. The contamination was also considered separately for DON, ZEA and the related emerging mycotoxins and fungal metabolites produced by the same *Fusarium* spp. for the two years because of the significant interaction between year and N rate. On average, the total DON (DON TOT) content, expressed as the sum of DON and its related masked forms, that is, deoxynivalenol-3-glucoside (DON-3-G), 3-acetyldeoxynivalenol (3-ADON) and 15-acetyldeoxynivalenol (15-ADON) (Figure 2), were much higher in 2013 (3690 µg kg^−1^) than in 2012 (366 µg kg^−1^).

When considering the average of the different N rates, DON accounted for about 49% of the DON TOT in 2012, DON-3-G for 41%, 3-ADON for 2%, and 15-ADON for 8%. On the other hand, DON was about 70% of the DON TOT in 2013, DON-3-G was 11%, 3-ADON was 4% and 15-ADON was 15%. Thus, DON-3-G showed a higher occurrence in the year with the lowest DON TOT content, i.e., in 2012. A significant increase (+12.5 times) in the DON TOT content was only recorded in 2013 for the N rate of 400 kg N ha^−1^, compared to the other N rates, while no differences were recorded in 2012. Although no differences were recorded between the N rates of 0, 100, 200 and 300 kg N ha^−1^ in 2013, a lower DON TOT content was observed for 100 and 200 kg N ha^−1^ (466 µg kg^−1^ on average) than that recorded for 0 and 300 kg N ha^−1^ (1777 µg kg^−1^ on average). 

A similar trend was recorded for the total ZEA (ZEA TOT) content, expressed as the sum of ZEA and its related masked forms, that is, zearalenone-4-sulfate (ZEA-4-S), α-zearalenol (α-ZEAol) and β-zearalenol (β-ZEAol) (Figure 3). Indeed, in the same way as for the DON TOT content, a higher contamination level was recorded in 2013 (114 µg kg^−1^) than in 2012 (39 µg kg^−1^). When considering the different average N rates, ZEA accounted for about 88% of the ZEA TOT in 2012, ZEA-4-S for 6%, α-ZEAol for 2%, and β-ZEAol for 4%. On the other hand, ZEA was about 76% of the ZEA TOT in 2013, ZEA-4-S was 17%, α-ZEAol was 2% and β-ZEAol was 5%. A significant increase (+9.5 times) in the ZEA TOT content was only recorded in 2013 for the N rate of 400 kg N ha^−1^, compared to the other N rates, while no differences were recorded in 2012, although the 400 kg N ha^−1^ rate presented the highest ZEA TOT content (135 µg kg^−1^). 

Although no differences were recorded between the N rates of 0, 100, 200 and 300 kg N ha^−1^ in 2013, a lower ZEA TOT content was observed for 100, 200 and 300 kg N ha^−1^ (on average 26 µg kg^−1^) than that recorded for 0 kg N ha^−1^ (89 µg kg^−1^). 

A similar trend was recorded for the effect of the N rate on the emerging mycotoxins and fungal metabolites detected in the maize grains (aurofusarin = AUR, butenolide = BUT, culmorin = CULM and nivalenol = NIV) and produced by the same *Fusarium* spp. section *Discolor* that produces DON and ZEA (Table 4). Indeed, the highest contents of all these metabolites were recorded in 2013 for the highest N rate of 400 kg N ha^−1^, which significantly differed from the other N rates considered in the trial, with increased average contents of +9.2% for AUR, 6.1% for BUT, 3.7% for CULM and 9.1% for NIV in comparison to all the others. No significant differences between the different N rates were observed in 2012 for any of the previously mentioned metabolites.

Equisetin (EQU), a *Fusarium* mycotoxin produced by species of the section *Roseum*, and T-2 and HT-2 toxins, which are *Fusarium* mycotoxins produced by species of the section *Sporotrichiella,* were only detected in traces in both years, and no significant differences were observed between the different N rates for either year.

### 2.6. Secalonic Acid D (SAD): An Aspergillus- and Penicillium-Derived Mycotoxin

SAD, an *Aspergillus*- and *Penicillium*-derived mycotoxin, was detected in both years, with a higher average content in 2012 (472 µg kg^−1^) than in 2013 (90 µg kg^−1^) (Figure 4). The SAD contamination was also considered separately for the two years because of the significant interaction between year and N rate. Although no differences were observed for the different N rates in 2012, the highest content was recorded for the 400 kg N ha^−1^ rate (992 µg kg^−1^) and was on average about +2.9 times higher than for the other N rates.

The higher N rates of 300 and 400 kg N ha^−1^ showed significantly higher SAD contents in 2013 (on average +12.3 times) than those recorded for the lower rates.

## 3. Discussion

This study was aimed at investigating, for the first time, the role of soil fertility, resulting from different N fertilization rates, on the contamination of regulated and emerging mycotoxins and fungal metabolites in maize grains at harvest in naturally-infected field trials carried out in the 2012–2013 growing seasons in a long-term experiment platform area (20-year period) in North-West Italy, where the maize received 5 different N rates, ranging from 0 to 400 kg N ha^−1^, each year.

Twenty-five of the most abundant mycotoxins and fungal metabolites were detected in both years and for almost all the considered N rates. The co-occurrence of regulated and emerging mycotoxins and fungal metabolites has already been reported in maize throughout Europe by several authors [17,18,19], thereby confirming the risks related to the presence of emerging mycotoxins and fungal metabolites in food and feeds. Therefore, the assessment of certain field conditions, such as N fertilization, which could lead to a higher contamination of these mycotoxins and fungal metabolites, is of fundamental importance to set up Good Agricultural Practices (GAP) with the aim of minimizing their occurrence.

As expected, since the application of N as a topdressing is a common practice to increase crop productivity [13], higher N rates (>200 kg N ha^−1^) significantly increased the grain yield, in comparison to low doses. On the other hand, as previously reported by Blandino et al. [14], the incidence and severity of fungal ear rot was generally higher in ears from plants fertilized with insufficient N. Similarly, Souza et al. [13] reported that the non-application of N (0 kg N ha^−1^) also resulted in higher grain infections of *Fusarium* spp., *Penicillium* spp. and yeasts.

Moreover, the obtained results underlined that the contamination of FBs, and of the other detected mycotoxins and fungal metabolites produced by *Fusarium* spp. section *Liseola,* was increased in poor soil fertility conditions. Souza et al. [13] and Blandino et al. [14] obtained similar results for FBs and reported that the non-application of N (0 kg N ha^−1^) resulted in a higher contamination of the kernels by FBs (FB_1_ and FB_2_). Ariño et al. [12], in a field study conducted in Brazil, reported that the use of N fertilization tended to increase the FB content of maize, and they showed a weak correlation (r = 0.46) between kilograms of N per hectare and FB level. Marocco et al. [15] reported that N fertilization resulted in significantly increased FB levels in maize grown in Italy, and they recorded that FBs in unfertilized and fertilized (270 kg N ha^−1^) fields increased by 99 and 70% in 2000 and 2001, respectively. Bocianowski et al. [16], in field trials conducted in Poland, also reported that an excess of N in the soil increased the frequency of grain infection with fungi of the *Fusarium* spp. genus. 

A deficiency or excess of plant-available N can lead to stress, thus increasing the susceptibility of the plants to attacks by pests and pathogens [20]. Furthermore, our results clearly show a positive correlation between N rate and grain yield, mainly as a consequence of an increase in the ear dimension. Therefore, a deficiency of N could result in small ears, characterized by a lower dilution of the kernels affected by ECB injuries and by healthy kernels less contaminated by FBs. This effect was, in fact, emphasized in our study in the year with the highest number of ECB injuries on maize ears (2012), thus confirming the key role of ECB in promoting *Fusarium* section *Liseola* infection in European maize growing areas [21] and the need for both a direct and indirect control of this pest as one of the main strategies adopted to minimize FBs and MON contamination [22,23].

The effect of N fertilization on BEA, BIK, FA, FUS, fusarin C and MON has never been reported before, and the obtained results have underlined that their occurrence is closely linked to FB contamination, as they are all produced by the same *Fusarium* spp. section *Liseola.* Moreover, the effect on N fertilization was found to be similar, and a higher risk of contamination was recorded under N deficiency conditions. This aspect should be considered in maize field management assessments in relation to their toxicological implications for humans and animals. Indeed, particular attention should be paid to fusarin C, due to its carcinogenic potential for humans [24], while Jonsson et al. [25] reported a high acute toxicity of MON in rats. Moreover, a recent review [26] underlined an interactive toxicity of MON and FB_1_. FA [27,28], BEA [29,30] and FUS [31,32] also showed toxic effects in humans and animals. Instead, there is a lack of toxicological data for BIK, and further studies are needed [33].

On the other hand, the obtained results underline that an overload of N fertilization was generally associated with higher DON and ZEA contaminations in maize kernels, as well as a higher risk of other emerging mycotoxins and fungal metabolites produced by *Fusarium* spp. sections *Discolor* and *Roseum*. Similarly, Blandino et al. [14] reported that a high N fertilizer application (> 300 kg N ha^−1^) significantly increased the ZEA content, whereas DON, when found, did not show a clear relationship with the N fertilization rate. 

Overall, contrary to what was observed for the FBs and the related emerging mycotoxins and fungal metabolites produced by *Fusarium* spp. section *Liseola*, DON, ZEA and the related emerging mycotoxins and fungal metabolites produced by *Fusarium* spp. section *Discolor* showed higher contamination levels in 2013 than in 2012, due to the large amount of rainfall recorded in July during maize flowering. In agreement with these results, other authors have also reported that the possibility of the development of *Fusarium* fungi on ears and grains increases for favorable weather conditions (higher temperatures and higher rainfall levels in summer, thus resulting in a longer growing season) [34]. Blandino et al. [35] reported that the highest DON and other *Fusarium* spp. section *Discolor* mycotoxin contaminations of maize grain occurred in growing seasons characterized by a high level of precipitation and lower temperatures in the period from maize flowering to maturation.

The effect of N fertilization on AUR, BUT, CULM, DON-3-G, 3- and 15-ADON, NIV, ZEA-4-S and α- and β-ZEAol has never previously been reported, and the obtained results underline that their occurrence is closely linked to DON and ZEA contamination, as they are all produced by the same *Fusarium* spp. section *Discolor.* Moreover, the effect of N fertilization was also comparable, with a higher risk of contamination being recorded for N overload conditions (>300 kg N ha^−1^). From the toxicological point of view, particular attention should be paid to DON-3-G and ZEA-4-S, which are the most commonly masked mycotoxins found in food and feeds. Their toxicological properties mainly involve the conversion of DON-3-G to DON and ZEA-S to ZEA by microbiota of the intestinal tract [36], thereby contributing to enhancing the total dietary exposure of individuals to the native forms DON and ZEA [37]. The α-ZEAol and β-ZEAol phase I plant metabolites of ZEA instead present a higher toxicity level and greater hyperestrogenic effects than ZEA, especially α-ZEAol [38]. Thus, all these modified forms should be taken into account for correct risk assessments and food safety [39]. Woelfingseder et al. [40] recently reported that CULM could act as a potentially co-occurring modulator of DON toxicokinetics in vivo and could therefore be classified not only as a secondary fungal metabolite but also as an “emerging mycotoxin”. AUR is considered a neglected mycotoxin [8,41], although it is known to induce oxidative stress, cytotoxicity and genotoxicity in human colon cells [41]. Finally, BUT could induce myocardial toxicity [42].

It is interesting to note that DON-3-G showed the highest contamination level and occurrence in the year with the lowest DON TOT content, i.e., in 2012, thus confirming an inverse relationship between the amount of DON and the DON-3-G/DON ratio, as was previously reported for wheat by Scarpino et al. [43]. Moreover, it is also important to underline that the maize grain dry-down could be affected by the soil fertility and N availability; an excess of N slows down the final ripening stage after physiological maturity [44], thereby resulting in a delayed harvest date, mostly in growing seasons characterized by higher levels of rainfall in summer. This could be a key element in explaining the increase in DON and its related mycotoxins and fungal metabolites that was observed in our study following the excess of N (400 kg N ha^−1^). The other detected mycotoxins and fungal metabolites, such as T-2 and HT-2 toxins (*Fusarium* spp. section *Sporotrichiella*) and EQU (*Fusarium* spp. section *Roseum*), were only detected in traces, and no significant differences were ever observed for the different N rates in either year.

Moreover, SAD, an *Aspergillus*-, *Penicillium*- or *Claviceps purpurea*-derived mycotoxin [45], was detected in both years, with a higher average content in 2012 than in 2013. A higher risk of SAD contamination was recorded for N overload conditions (>300 kg N ha^−1^). Although the effect of N fertilization on SAD contamination has never been reported before, Blandino et al. [14], contrary to this trend, found a negative correlation between N rate and aflatoxin B_1_, the main *Aspergillus* mycotoxin, when the climatic conditions during ripening favored this mycotoxin. SAD is a highly toxic, teratogenic, and weakly mutagenic mycotoxin and is a common contaminant in the United States. Because of its teratogenic effects, SAD affects pregnant mice and their progeny and acts in a dose-dependent manner that leads to neurotoxicity [45].

Overall, the production of secondary metabolites is influenced by global regulators which directly or indirectly activate the respective genes or gene clusters. The global regulators respond to a variety of abiotic components including pH, light, temperature, water activity and nutrient availability [46]. N is an essential requirement for growth, and the ability to metabolize a wide variety of N sources enables fungi to colonize different environmental niches and survive nutrient limitations. However, despite the progress made in studying N regulation of secondary metabolism, the molecular mode of action and possible interactions between and cross-talks with the regulators are not well understood. Because of the importance of N availability in regulating secondary metabolism, fundamental studies are needed [47].

In conclusion, the results obtained in the present field experiment suggest that a balanced application of N fertilizer, that in the considered production situation was a comprise between 200 and 300 kg N ha^−1^, according to data reported by Zavattaro et al. [48], generally seems to guarantee a lower risk of mycotoxin contamination and may therefore represent the best solution to minimize the overall contamination of both regulated and emerging mycotoxins and fungal metabolites.

## 4. Materials and Methods

### 4.1. Experimental Design

The effect of nutritional stresses on regulated, modified and emerging mycotoxins was studied in the 2012 and 2013 growing seasons on selected plots of the long-term experiment conducted at Tetto Frati (44°53′ N; 7°41′ E; 232 m a.s.l.) by the University of Turin (NW Italy). The long-term experiment, which was started in 1992, has been described by Grignani et al. [49] and Zavattaro et al. [48], and it has involved comparisons of different cropping systems based on maize and five N fertilization levels in a complete randomized block with three replicates on 75 m^2^ plots. The present study has only been carried out on the continuous maize for grain system at 5 mineral fertilizer rates, that is, 0, 100, 200, 300 and 400 kg of N ha^−1^, supplied each year as granular urea (46%), partly distributed a few days before maize sowing and partly top-dressed and incorporated through ridging.

Briefly, the soil is a loam and is classified as *Typic Udifluvent*. The ploughed layer (0–30 cm) contains 48.2, 44.3 and 7.5% of sand, silt and clay, respectively, and has a sub-alkaline soil pH (8.1 measured in water at 1:2.5 *w*/*v*) and a low Cation Exchange Capacity (10.1 cmol(+) kg^−1^). With the exception of the N fertilization rate, the crop was managed similarly in all the plots, according to the conventional management of maize in the growing area. The soil was plowed in autumn and the maize residues (stalks, cobs and bracts) were incorporated. Mechanical maize seeding was carried out on 27 April 2012 and on 9 May 2013, using the Monsanto Dekalb DKC6677 hybrid (FAO maturity class 600, 130 days relative to maturity). Distances of 0.16 m and 0.75 m were adopted between the plants and the plant rows, respectively, thus providing a crop density of 8.3 plant m^−2^. The plots were weeded during pre- and post-emergence. Irrigation was carried out by means of the sprinkling method, and approximately 100 mm was supplied per year in order to avoid drought stress. Weeds were controlled with mesotrione, metolachlor and terbutilazine in pre-emergence and sulcotrione and nicosulfuron in post- emergence. No foliar insecticide or fungicide was applied during the cultivation. 

The climate at the site is temperate sub-continental, with two main rainy periods in spring and autumn. The daily temperatures and precipitations were measured at a meteorological station located in the experimental platform area. The accumulated growing degree days (GDDs) for maize were calculated considering 10 °C as the minimum base temperature.

### 4.2. Grain Yield

Ears were collected by hand at harvest maturity from 4.5 m^2^ in each plot to quantify the grain yield and to obtain a representative sample for mycotoxin analysis. All the plots were harvested on 19 September 2012 and 23 October 2013. 

All the collected ears were shelled using a mechanical sheller. The kernels from each plot were mixed thoroughly to obtain a uniform sample, and grain moisture was analyzed using a Dickey-John GAC2100 grain analyzer (DICKEY-john, Auburn, IL, USA). The grain yields were adjusted to a 14% moisture content. A 5 kg sub-sample was taken for the mycotoxin analyses and dried at 60 °C for 72 hours in order to reduce the kernel moisture content to 10% before the milling operation.

### 4.3. ECB and Fungal Ear Rot Symptoms and Fungal Infection

A sub-sample of 30 ears was used, after removing the husks, to evaluate ECB and fungal ear rot severity at harvest, calculated as the percentage of kernels per ear with symptoms according to the procedure reported by Blandino et al. (2009) [23].

### 4.4. Multi-Mycotoxin LC-MS/MS Analysis

The extraction phase was described in detail by Sulyok et al. in 2006 [50]. The chromatographic and mass spectrometric parameters of the investigated analytes were described by Malachova et al. in 2014 [51]. 

### 4.5. Statistical Analysis

The Kolmogorov-Smirnov normality test and the Levene test were carried out to verify the normal distribution and homogeneity of variances, respectively. Since the interaction between the N rate and year was significant, an analysis of the variance (ANOVA) was performed separately for each year with the N rate as an independent factor for the fungal ear rot incidence and severity, ECB incidence and severity and mycotoxin contamination. Multiple comparison tests were performed, with the Ryan–Einot–Gabriel–Welsh F (REGW-F) test, on the treatment means. A statistical data analysis was carried out with the SPSS software package, version 24.0.

## Figures and Tables

**Figure 1 toxins-14-00448-f001:**
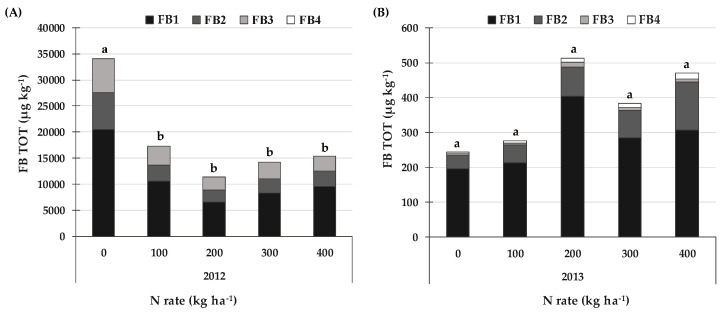
Effect of the N rate on fumonisin B (FBs = sum of fumonisin B_1_, FB_1_; fumonisin B_2_, FB_2_; fumonisin B_3_, FB_3_ and fumonisin B_4_, FB_4_) contamination in field experiments carried out in 2012 (**A**) and 2013 (**B**) in North-West Italy. Bars with different letters above are significantly different (*p*-value < 0.05), according to the REGW-F test.

**Figure 2 toxins-14-00448-f002:**
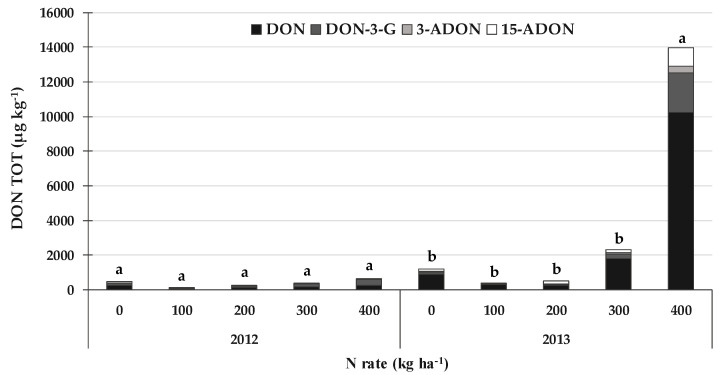
Effect of the N rate on the total deoxynivalenol (DON TOT = sum of deoxynivalenol, DON; deoxynivalenol-3-glucoside, DON-3-G; 3-acetyldeoxynivalenol, 3-ADON; and 15-acetyldeoxynivalenol, 15-ADON) contamination in field experiments carried out in 2012 and 2013 in North-West Italy. Bars with different letters above are significantly different (*p*-value < 0.05), according to the REGW-F test.

**Figure 3 toxins-14-00448-f003:**
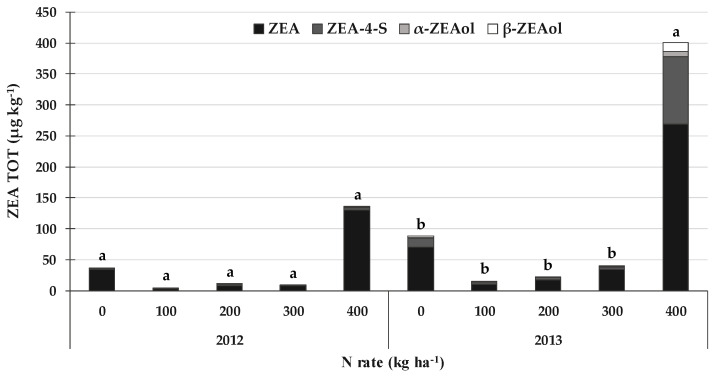
Effect of the N rate on the total zearalenone (ZEA TOT = sum of zearalenone, ZEA; zearalenone-4-sulfate, ZEA-4-S; α-zearalenol, α-ZEAol; and β-zearalenol, β-ZEAol) contamination in field experiments carried out in 2012 and 2013 in North-West Italy. Bars with different letters above are significantly different (*p*-value < 0.05), according to the REGW-F test.

**Figure 4 toxins-14-00448-f004:**
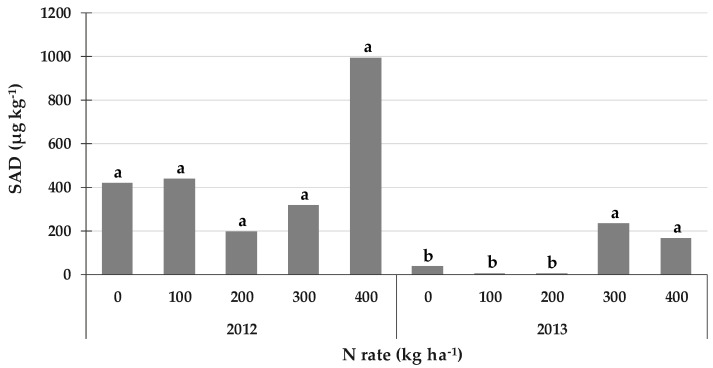
Effect of the N rate on secalonic acid D (SAD) contamination in field experiments carried out in 2012 and 2013 in North-West Italy. Bars with different letters above are significantly different (*p*-value < 0.05), according to the REGW-F test.

**Table 1 toxins-14-00448-t001:** Total rainfall, rainy days and growing degree days (GDDs) from April to October in 2012 and in 2013 at the research site.

Year	Month	Rainfall	Rainy Days n°	GDDs *^a^*
		mm		Σ °C-day
2012	April	148	13	112
	May	147	6	242
	June	19	2	374
	July	37	4	415
	August	46	2	424
	September	50	5	264
	October	53	4	169
	Sowing–Harvest *^b^*	322	19	1673
2013	April	144	9	145
	May	147	11	210
	June	35	2	325
	July	137	7	430
	August	59	3	390
	September	14	1	271
	October	71	9	145
	Sowing–Harvest *^b^*	426	27	1657

*^a^* Accumulated growing degree days for each experiment using a 10 °C base value. *^b^* The sowing–harvest periods were: 27 April–20 September and 9 May–22 October for 2012 and 2013, respectively.

**Table 2 toxins-14-00448-t002:** Effect of the N rate on the grain yield, incidence and severity of fungal ear rot and European Corn Borer (ECB) for field experiments carried out in North-West Italy in 2012 and 2013.

Year	N Rate	Grain Yield	Ear Rot Incidence *^a^*	Ear Rot Severity *^b^*	ECB Incidence *^c^*	ECB Severity *^d^*
	(kg ha^−1^)	(t ha^−1^)	(%)	(%)	(%)	(%)
2012	0	6.7 c	96.7 a	18.6 a	96.7 a	17.3 a
	100	9.3 b	81.7 a	5.3 b	80.0 a	10.2 b
	200	11.6 a	95.0 a	5.3 b	85.0 a	8.6 b
	300	12.4 a	95.0 a	6.0 b	91.7 a	10.0 b
	400	13.1 a	91.7 a	6.4 b	86.7 a	9.5 b
	*p*-value	<0.001	0.575	0.002	0.215	0.010
2013	0	10.9 c	25.0 a	5.9 a	6.7 a	1.0 a
	100	13.3 b	26.9 a	2.9 a	23.6 a	1.6 a
	200	14.8 a	22.1 a	2.0 a	11.9 a	1.3 a
	300	14.5 ab	43.3 a	3.7 a	31.7 a	2.3 a
	400	14.8 a	42.9 a	9.8 a	27.5 a	3.0 a
	*p*-value	<0.001	0.490	0.302	0.473	0.701

Means followed by different letters are significantly different (the level of significance of the *p*-value is reported in the table), according to the REGW-F test. The reported data for each year and N rate are the average of 3 replications. *^a^* Fungal ear rot incidence was calculated as the percentage of ears with symptoms, considering 3 replications of 30 ears each. *^b^* Fungal ear rot severity was calculated as the mean percentage of kernels with symptoms per ear, considering 3 replications of 30 ears each. *^c^* ECB incidence was calculated as the percentage of ears with symptoms, considering 3 replications of 30 ears each. *^d^* ECB severity was calculated as the mean percentage of kernels with symptoms per ear, considering 3 replications of 30 ears each.

**Table 3 toxins-14-00448-t003:** Effect of the N rate on the contamination of beauvericin (BEA), bikaverin (BIK), fusaric acid (FA), fusaproliferin (FUS), fusarin C and moniliformin (MON) in field experiments carried out in North-West Italy in 2012 and 2013.

Year	N Rate	BEA	BIK	FA	FUS	Fusarin C	MON
	(kg ha^−1^)	(µg kg^−1^)	(µg kg^−1^)	(µg kg^−1^)	(µg kg^−1^)	(µg kg^−1^)	(µg kg^−1^)
2012	0	891 a	969 a	1305 a	5810 a	<LOD*^a^*	1937 a
	100	280 a	486 ab	266 b	2409 b	<LOD	1554 ab
	200	347 a	360 b	245 b	2563 b	<LOD	1061 b
	300	688 a	466 b	323 b	3183 b	<LOD	1575 ab
	400	1093 a	496 ab	246 b	2583 b	<LOD	1926 a
	*p*-value	0.067	0.040	0.001	0.005	-	0.023
2013	0	9 a	47 a	207 a	409 a	<LOD a	75 a
	100	3 a	62 a	140 a	26 a	<LOD a	18 a
	200	14 a	97 a	277 a	221 a	39 a	27 a
	300	19 a	193 a	247 a	73 a	91 a	99 a
	400	9 a	109 a	269 a	92 a	142 a	53 a
	*p*-value	0.714	0.095	0.777	0.345	0.134	0.427

Means followed by different letters are significantly different (the level of significance of the *p*-value is reported in the table), according to the REGW-F test. The reported data are the average of 3 replications. *^a^* LOD of fusarin C = 4.8 µg kg^−1^.

**Table 4 toxins-14-00448-t004:** Effect of the N rate on the contamination of aurofusarin (AUR), butenolide (BUT), culmorin (CULM), equisetin (EQU), nivalenol (NIV) T-2 and HT-2 toxins in field experiments carried out in North-West Italy in 2012 and 2013.

Year	N Rate	AUR	BUT	CULM	EQU	NIV	T-2 Toxin	HT-2 Toxin
	(kg ha^−1^)	(µg kg^−1^)	(µg kg^−1^)	(µg kg^−1^)	(µg kg^−1^)	(µg kg^−1^)	(µg kg^−1^)	(µg kg^−1^)
2012	0	468 a	56 a	171 a	7 a	<LOD *^a^* a	4 a	12 a
	100	78 a	53 a	22 a	205 a	2 a	<LOD *^b^* a	<LOD *^c^* a
	200	58 a	33 a	72 a	42 a	3 a	3 a	9 a
	300	265 a	39 a	39 a	171 a	<LOD a	5 a	9 a
	400	606 a	36 a	73 a	209 a	<LOD a	9 a	7 a
	*p*-value	0.251	0.791	0.107	0.204	0.555	0.767	0.873
2013	0	1171 b	131 b	885 b	3 a	3 b	0.9 a	<LOD a
	100	588 b	70 b	1910 b	5 a	<LOD b	<LOD a	<LOD a
	200	2822 b	111 b	1713 b	1 a	3 b	<LOD a	<LOD a
	300	1469 b	119 b	1970 b	32 a	2 b	<LOD a	<LOD a
	400	13876 a	656 a	6004 a	41 a	23 a	3 a	4 a
	*p*-value	<0.001	0.039	0.038	0.490	<0.001	0.507	0.237

Means followed by different letters are significantly different (the level of significance of the *p*-value is reported in the table), according to the REGW-F test. The reported data are the average of 3 replications. *^a^* LOD of nivalenol = 1.2 µg kg^−1^. *^b^* LOD of T-2 toxin = 0.8 µg kg^−1^. *^c^* LOD of HT-2 toxin = 3.2 µg kg^−1^.

## Data Availability

The data that support the findings of this study are available from the corresponding author, upon reasonable request.

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
