# Peer review of "The Role of Nitrogen Fertilization on the Occurrence of Regulated, Modified and Emerging Mycotoxins and Fungal Metabolites in Maize Kernels"

_toxins, 2022, doi:10.3390/toxins14070448_

Round 1

Reviewer 1 Report

In chapter 4.4. Multi Mycotoxin LC-MS/MS analysis, works of Sulyok et al. (2006) and Malachova et al. (2014) related to Chromatography are citated. I think that in that part, the extraction method, column, type of instrument and LC-MS/MS parameters should be briefly stated, regardless of the citated works.

Author Response

We thank the Reviewer for the positive comment. The suggested addition of the information relating to chapter 4.4 has been done.

Reviewer 2 Report

The authors present interesting data on mycotoxins present in maize kernels and put these into context to further metadata directly connected to the area where these kernels were produced. The study is well designed and the data presented are ok. The reviewer did not find any data on standard deviation and the dilute and shut method is not very sensitive, it's sad the researchers did not set up a proper detection multi-method. The authors should take into consideration changing the title to a more general title since they did not really test the influences on the fungus and its metabolite production. Many more factors could be involved and it can not be said for sure where this is really the impact that causes more toxin production. In addition, the abstract should be rewritten since reading the article and the abstract confuses the reader. Finally, in the discussion, the authors could also include research on nutritional factors influencing the secondary metabolism of Fusarium spp. (e.g., W.-B. Shim et al. 1994) to further ratify their findings. The method could be described in more detail but is ok.

Author Response

Comments and Suggestions for Authors

The authors present interesting data on mycotoxins present in maize kernels and put these into context to further metadata directly connected to the area where these kernels were produced. The study is well designed and the data presented are ok.

We thank the Reviewer for the positive comment.

The reviewer did not find any data on standard deviation and the dilute and shut method is not very sensitive, it's sad the researchers did not set up a proper detection multi-method.

All the information related to the performances of the method was reported in Sulyok et al., 2006 and Malachova et al., 2014, all cited in the manuscript.

The authors should take into consideration changing the title to a more general title since they did not really test the influences on the fungus and its metabolite production. Many more factors could be involved and it can not be said for sure where this is really the impact that causes more toxin production.

In the title and in the text we referred to emerging fungal metabolites not for the fact that we went to evaluate the metabolism of fungi and the influence of nitrogen fertilization on this aspect. The term emerging fungal metabolites referred to secondary fungal metabolites with low or unknown toxicological data/information, such as aurofusarin, culmorin, butenolide etc. Indeed, as reported by Jestoi et al., (2008), Fusarium spp. are also capable of producing other toxic secondary metabolites, the so-called emerging mycotoxins such as fusaproliferin, beauvericin, enniatins, and moniliformin, for which, even if only partially, their toxic effects for animals and humans have already been described and evaluated.

As far as the factors that could be involved in the mycotoxin production is concerned, we agree with the reviewer that exists several interacting factors that influence the production of mycotoxins by the different producing fungi, but in our study, the other agronomical factors have been kept constant in the different years, while a focus on soil fertility, resulting from different nitrogen fertilization rates, was carried out to evaluate its effect under the same other growth conditions in the field.

According also to the suggestion of reviewer 4, the title has been changed to “The role of nitrogen fertilization on the occurrence of regulated, modified and emerging mycotoxins and fungal metabolites in maize kernels”.

In addition, the abstract should be rewritten since reading the article and the abstract confuses the reader.

As suggested by the reviewer, and in order to not to confuse the readers, the following sentence has been deleted by the abstract: “Therefore, the assessment of the field conditions that lead to increases in contamination is necessary to set up Good Agricultural Practices that minimize their occurrence.” According also to the other reviewer suggestions, the abstract has been revised.

Finally, in the discussion, the authors could also include research on nutritional factors influencing the secondary metabolism of Fusarium spp. (e.g., W.-B. Shim et al. 1994) to further ratify their findings.

As suggested by the reviewer a paragraph in the discussion section has been added in order to describe the nutritional factors influencing the second metabolism of fungi (L411-420).

The method could be described in more detail but is ok.

The suggested addition of the information relating to the analytical method has been done.

Reviewer 3 Report

Two-years field experiment is not really allows us to make significant and real deductions. Too many factors are there affecting the measured values. It is statistically unmanageable, as the authors also found and treated the two years' results separately in most cases.

Moreover, for N supplementation, recommending 200 kg ha-1, I think it depends on the soil type, what we will find as recommendable N content. Total soil N is also needed to be measured and considered as basis in the deduction. 

Author Response

Two-years field experiment is not really allows us to make significant and real deductions. Too many factors are there affecting the measured values. It is statistically unmanageable, as the authors also found and treated the two years' results separately in most cases.

The need to elaborate the 2 growing season separately, was related to the different occurrence of regulated, modified and emerging mycotoxins detected as a consequence of the clear different meteorological trends recorded during the 2 growing seasons.

The data have been statistically analyzed, since all the treatments have been biological replicated, and the analysis of the effect of N fertilization carried out separately for year could allow a clearer quantification of its role on the possible contamination in maize grain.

Although the replication of the field experiment several growing seasons could be preferable, a 2-year study allows in our opinion to begin to understand and confirm the observed effects. Indeed, the data collected in the 2-year study were clear and in line with other studies carried out exclusively on regulated mycotoxin (FBs and DON).

We think that the present work is of interest for Toxins readers, since investigate the role of soil fertility and N fertilization, and related stress, also on modified and emerging mycotoxins and other fungal metabolites, reporting data of 2 growing seasons characterized by clear different meteorological conditions that allow to observe the effect on a wide range of toxic compounds. 

Moreover, for N supplementation, recommending 200 kg ha-1, I think it depends on the soil type, what we will find as recommendable N content. Total soil N is also needed to be measured and considered as basis in the deduction. 

The reviewer is right, the recommendation of the N fertilization is an agronomical aspect strongly related to the pedo-climatic, agronomic, and crop productivity potential, thus it is incorrect state a general value.

All the information related to the plant uptakes, N soil content (also in soil water) and N balance for the compared N fertilization treatment are reported in the manuscript Grignani et al. [50] and Zavattaro et al. [49], which are cited in the manuscript and which describe in deep the agronomical information of the experiment.

The conclusion (also in the abstract) has been change to “In conclusion, the results obtained in the present field experiment suggested that a balanced application of N fertilizer, in accordance with maize uptake, that in the considered production situation were comprise between 200 and 300 kg N ha-1, according to data reported by Zavattaro et al. [49], generally seems to guarantee a lower risk of mycotoxin contamination and may therefore represent the best solution to minimize the overall contamination of both regulated and emerging mycotoxins and fungal metabolites.”

Reviewer 4 Report

General comments: In this present study, the authors evaluated the effect of nitrogen fertilization on mycotoxins production in maize. The study was very meaningful for mycotoxins control before harvest, which will be helpful for developing appropriate field management. However, there are some key problems that should be clarified before being considered for publication in this paper. 

Title: the objective of this study is to evaluate the effect of nitrogen fertilization on mycotoxin production in maize. I suggest the title should be correct as “the role of nitrogen fertilization on the occurrence of regulated, modified and emerging mycotoxins and fungal metabolites in maize kernels”.

What did the authors mean for “fungal metabolites”? As mentioned in this manuscript, fungal metabolites may belong to emerging mycotoxins?

Line 15:Fusarium spp. of the Liseola section” should be “Fusarium section Liseola

Line 23-24: Keywords: the keywords cannot clarify the topic of this study!!!

Line 42: What are the fungal metabolites?

Line 84:  The authors should point out which growing stage was the key period affecting mycotoxins production. Were the climatic temperatures in the field trial site recorded? In addition to the rainfall, the temperature is another very important factor affecting the occurrence of mycotoxins.

Line 136: As mentioned in the manuscript, the ear rot and ECB incidences were never affected by the N rate, but the severity of the ear rot resulted in a significantly higher for a zero N rate than other N input in 2012. What was the relationship between ear rot and mycotoxin contamination? Which fungi are the major causes of maize ear rot? Besides, as shown in Figure2, Figure3, and Table 4, the contamination levels of deoxynivalenol and its derivatives, zearalenone and its derivatives were higher in 2013 than that in 2012. However, the ear rot in 2013 seemed not serious. Could the authors explain the possible reasons?

Line 154: Figure 1: I suggested that Figure 1 was divided into Figure 1a and Figure 1b in order to check the figure information better.

Line 195: I’m very curious that no different significance between N rates was observed for BEA (the BEA levels ranged from 280-1093 µg/kg). Please check the statistical analysis in table 3 again. Also, Figure 2 and Figure 3!!

Line 302: the authors did not carry out the pathogen identification in this study. How did you confirm that FBs were produced by Fusarium section Liseola?, even though it mainly produced FBs.

Line 443: The representative samples were collected from 4.5 m2 in the central rows of each plot for mycotoxin analysis. Was this a standard procedure for sample collection in Italy?

Line 470: In this study, the mycotoxins analysis was very important data. The authors should describe this part in detail. After reading this paper, I even don’t know how many mycotoxins were tested, and which types of mycotoxins were analyzed. In addition, the accuracy of the method was verified by participating in proficiency testing schemes organized by BIPEA. In this testing scheme, were all the toxins in this study covered? If not, how did the authors ensure the method reliability?

Author Response

General comments: In this present study, the authors evaluated the effect of nitrogen fertilization on mycotoxins production in maize. The study was very meaningful for mycotoxins control before harvest, which will be helpful for developing appropriate field management. However, there are some key problems that should be clarified before being considered for publication in this paper.

We thank the Reviewer for the valuable comments that will allow us to improve the manuscript.

Title: the objective of this study is to evaluate the effect of nitrogen fertilization on mycotoxin production in maize. I suggest the title should be correct as “the role of nitrogen fertilization on the occurrence of regulated, modified and emerging mycotoxins and fungal metabolites in maize kernels”.

The title has been modify as suggested by the reviewer.

What did the authors mean for “fungal metabolites”? As mentioned in this manuscript, fungal metabolites may belong to emerging mycotoxins?

The term emerging fungal metabolites referred to secondary fungal metabolites with low or unknown toxicological data/information, such as aurofusarin, culmorin, butenolide etc. Indeed, as reported by Jestoi et al. (2008) Fusarium spp. are also capable of producing other toxic secondary metabolites, the so-called emerging mycotoxins such as fusaproliferin, beauvericin, enniatins, and moniliformin, for which, even if only partially, their toxic effects for animals and humans have already been described and evaluated. In our work we have considered both fungal metabolites for which is confirmed the toxic activity (regulated, modified and emerging mycotoxin) and other fungal metabolites, for which the toxicological profile need to be studied in deep (but that for this reason could not be considered mycotoxin at the moment).

Line 15: “Fusarium spp. of the Liseola section” should be “Fusarium section Liseola”.

The suggested modification has been made throughout the text.

Line 23-24: Keywords: the keywords cannot clarify the topic of this study!!!

We considered appropriate to choose as keywords other words not already present in the title that would allow the readers to improve the search for the article. Since the focus of the manuscript are the regulated, modified and emerging mycotoxins, their specific name have been inserted as keywords.

Line 42: What are the fungal metabolites?

The secondary fungal metabolites are all the metabolites produced by fungi, for which low, unknown or scarce information are present in literature about their toxicological effects on humans and animals and cannot be called mycotoxins, such as aurofusarin, culmorin, butenolide.

Line 84:  The authors should point out which growing stage was the key period affecting mycotoxins production. Were the climatic temperatures in the field trial site recorded? In addition to the rainfall, the temperature is another very important factor affecting the occurrence of mycotoxins.

We addressed the influence of meteo-climatic aspects (rainfall and temperature) on mycotoxin contamination in the discussions section at L350-361. Moreover, as reported at L455-456 the daily temperatures (reported as GDDs) and precipitations were measured at a meteorological station located in the experimental platform area. Both rainfalls and temperatures, during all the growing seasons since they have a direct influence on crop development, are key factors in addressing the mycotoxin contamination recorded in each growing season of the field study.

Line 136: As mentioned in the manuscript, the ear rot and ECB incidences were never affected by the N rate, but the severity of the ear rot resulted in a significantly higher for a zero N rate than other N input in 2012. What was the relationship between ear rot and mycotoxin contamination? Which fungi are the major causes of maize ear rot? Besides, as shown in Figure2, Figure3, and Table 4, the contamination levels of deoxynivalenol and its derivatives, zearalenone and its derivatives were higher in 2013 than that in 2012. However, the ear rot in 2013 seemed not serious. Could the authors explain the possible reasons?

Overall, the ear rot incidence and severity measured in the present study takes into account all the fungi involved in determining ear rot, without distinguishing the causal agent.

As reported by Gaikpa and Miedaner, 2019: “About 19 Fusarium species were reported to induce ear rots (ER) in maize (Mesterházy et al. 2012) producing a large number of chemically very different mycotoxins (Logrieco et al. 2002). The two major Fusarium species in maize in temperate areas are, however, Fusarium graminearum and F. verticillioides.

Fusarium graminearum (Schwabe, teleomorph: Gibberella zeae) causes Gibberella ear rot (GER, maize red ear rot) as well as stalk rot (Pè et al. 1993) and produces primarily the mycotoxins deoxynivalenol and zearalenone in infected grains. High infections occur when the weather is cool and wet at early silking (Mouton 2014; Wise et al. 2016).

Fusarium ear rot (FER, maize pink ear rot) is mainly caused by Fusarium verticillioides (Sacc.) Nirenberg (syn. F. moniliforme Sheldon), but also F. proliferatum, F. subglutinans, F. temperatum sp. nov. and other Fusarium species may contribute to this disease (Logrieco et al. 2002). F. verticillioides causes ER in a wide range of environments throughout the world. However, severe infections occur under warm environments. Additionally, damages by insects or hail may boost infections tremendously because they create wounds that can readily be colonized by the fungi. In maize pink ear rot, which is mainly caused by F. verticillioides, there is increasing evidence of the wide occurrence of fumonisin B1 (Logrieco et al. 2012).”

Although the GER and FER result in different visual symptoms on maize ear, it is possible that both fungal agents could be present at the same time and only the fungal identification could permit an accurate quantification of the species involved.

Furthermore, in our study the quantification of Ear rot incidence and severity (and ECB damage) could provide a first interpretation of the results: in other words, understand if the increase of mycotoxin contamination in a certain growing season is first related to an increase in fungal development during maize ripening and/or to an increase in the toxigenic capacity.

Considering the above, in 2012 the higher ECB severity has influenced the higher Ear Rot severity, mainly due to FER, caused by F. verticillioides, leading to a higher FBs contamination, while in 2013, despite the overall lower ear rot severity (mainly related to higher grain yield, thus to higher dilution of the moldy portion in a bigger health maize ear) it could be mainly associated and caused by F. graminearum (GER), due also to the meteorological condition occurred in 2013 (abundant rainfall during maize flowering, and averaged lower temperature), which led to a later harvest than in 2012. This hypothesis is also confirmed by an extreme low contamination by fumonisins recorded in 2012, which suggests a low presence of FER.

Finally, it is necessary to consider that the fungal strain involved could determine a strongly different toxigenic capacity, therefore, as is well reported in the literature, fungal symptoms similar at harvest are not necessarily comparable as regards the mycotoxin content.

Line 154: Figure 1: I suggested that Figure 1 was divided into Figure 1a and Figure 1b in order to check the figure information better.

As suggested by the reviewer the Figure 1 has been divided into Figure 1A and Figure 1B in order to check the figure information better.

Line 195: I’m very curious that no different significance between N rates was observed for BEA (the BEA levels ranged from 280-1093 µg/kg). Please check the statistical analysis in table 3 again. Also, Figure 2 and Figure 3!!

We have checked the statistical analysis for BEA and in 2012 the p-value is 0.067 and therefore, even if slightly higher than 0.050, we cannot consider the different nitrogen doses statistically different. This is due to the biological variability present in the field (replications of treatments in different plots), that inevitably led to higher variation of certain values in the replicates. We have also checked the Figure 2 and Figure 3.

Line 302: the authors did not carry out the pathogen identification in this study. How did you confirm that FBs were produced by Fusarium section Liseola?, even though it mainly produced FBs.

Although we have not carried out the pathogen identification, we have reported the information presented in literature, which confirm what we have described. This allowed us to combine and describe some mycotoxins that showed the same behavior most likely because they were mainly produced by the same fungal species, as well known and reported in literature.

Line 443: The representative samples were collected from 4.5 m2 in the central rows of each plot for mycotoxin analysis. Was this a standard procedure for sample collection in Italy?

As reported in the M&M section, a careful procedure of sampling, already described and used in several manuscripts on maize, was carried out in order to obtain a representative sample from each plot.

All maize ears were collected, by hand, at harvest maturity from 4.5 m2 in the two central rows of each plot to obtain a representative sample for mycotoxin analysis, in accordance to the dimension of the experimental plot. The kernels from each plot were mixed thoroughly to obtain a uniform sample. A large sub-sample (5 kg) was taken for the mycotoxin analyses and dried at 60°C for 72 hours, in order to reduce the kernel moisture content to 10% before the milling operation. Dried maize grain samples (5 kg for each plot) were all ground using a ZM 200 Ultra Centrifugal Mill (Retsch GmbH, Haan, Germany) fitted with a 1 mm aperture sieve and the resulting whole meal, after a careful mixing operation, was used directly for the extraction.

Line 470: In this study, the mycotoxins analysis was very important data. The authors should describe this part in detail. After reading this paper, I even don’t know how many mycotoxins were tested, and which types of mycotoxins were analyzed. In addition, the accuracy of the method was verified by participating in proficiency testing schemes organized by BIPEA. In this testing scheme, were all the toxins in this study covered? If not, how did the authors ensure the method reliability?

The suggested addition of the information relating to the analytical method described in chapter 4.4 has been done. The other information that could be of reviewer interest could be found in Sulyok et al., 2006 and Malachova et al., 2014. Overall, applying the analytical method 295 bacterial and fungal metabolites, including all regulated mycotoxins, have been investigated in the samples, but we only presented the mycotoxins and fungal metabolites detected in our samples.

Round 2

Reviewer 3 Report

Thank you for your answer. I really appreciate your effort to make the manuscript better. I acknowledge your opinion on statistically separate the 2-years experiment, and hope you will have a third-year yield to analyse the connections of secondary metabolite production and climate. Considering the separated statistical analysis I accept the manusrcipt. 

Reviewer 4 Report

I recommend accepting this paper in toxins